# PARROT: Prediction of enzyme abundances using protein-constrained metabolic models

**Mauricio Alexander de Moura Ferreira**[1], **Wendel Batista da Silveira**[1], **Zoran Nikoloski**[2,3]*

1 Department of Microbiology, Federal University of Viçosa, Viçosa, Minas Gerais, Brazil, 2 Bioinformatics, Institute of Biochemistry and Biology, University of Potsdam, Potsdam, Germany, 3 Systems Biology and Mathematical Modelling, Max Planck Institute of Molecular Plant Physiology, Potsdam, Germany

* zoran.nikoloski@uni-potsdam.de

**Data Availability Statement:** All data and code are publicly available in the GitHub repository: https://github.com/mauricioamf/PARROT.

**Funding:** This study was financed in part by the Coordenação de Aperfeiçoamento de Pessoal de Nível Superior – Brasil (CAPES) – Finance Code

## Abstract

Protein allocation determines the activity of cellular pathways and affects growth across all organisms. Therefore, different experimental and machine learning approaches have been developed to quantify and predict protein abundance and how they are allocated to different cellular functions, respectively. Yet, despite advances in protein quantification, it remains challenging to predict condition-specific allocation of enzymes in metabolic networks. Here, using protein-constrained metabolic models, we propose a family of constrained-based approaches, termed PARROT, to predict how much of each enzyme is used based on the principle of minimizing the difference between a reference and an alternative growth condition. To this end, PARROT variants model the minimization of enzyme reallocation using four different (combinations of) distance functions. We demonstrate that the PARROT variant that minimizes the Manhattan distance between the enzyme allocation of a reference and an alternative condition outperforms existing approaches based on the parsimonious distribution of fluxes or enzymes for both *Escherichia coli* and *Saccharomyces cerevisiae*. Further, we show that the combined minimization of flux and enzyme allocation adjustment leads to inconsistent predictions. Together, our findings indicate that minimization of protein allocation rather than flux redistribution is a governing principle determining steady-state pathway activity for microorganism grown in alternative growth conditions.

## Author summary

Protein allocation determines the activity of cells and affects diverse traits across all organisms. However, prediction of protein allocation, particularly for conditions that do not result at optimal growth and physiology, remains a very challenging problem. In this study, we present an approach called PARROT to predict how cells allocate their proteins in different conditions. We tested different variants of PARROT by considering different objectives within a constraint-based formulation and by how much resource allocation information is used to guide predictions. We found that minimizing adjustments in protein allocation, rather than flux phenotypes, is a key principle that microorganisms use under alternative growth conditions. By integrating this principle into our approaches

001 to MAMF. The funders had no role in study design, data collection and analysis, decision to publish, or preparation of the manuscript.

**Competing interests:** The authors have declared that no competing interests exist.

and leveraging quantitative proteomics data, PARROT provides more accurate predictions of protein allocation in unseen conditions in comparison to existing contenders. Therefore, PARROT can help in advancing our understanding of protein allocation under different conditions and its physiological implications. Further, we can gain valuable insights into cellular responses and adaptive strategies across different environments.

## Introduction

Constraint-based approaches have been employed to simulate and predict phenotypes based on genome-scale metabolic models (GEMs) [1]. While already useful for predicting a wide range of phenotypes, the predictive performance of GEMs has been further improved by integrating protein constraints, such as: enzyme catalytic rates and the allocation of enzyme abundances across reactions [2,3]. Enzyme abundances are central to metabolic function, since they impact the rate of reactions and regulation of pathways, with implications to biotechnology and medicine [4]. These protein-constrained GEMs (pcGEMs) have been used to predict complex phenotypes, such as the overflow metabolism, in which fermentation predominates over respiration when microorganisms grow in high sugar concentrations [3,5], and diauxic growth, when multiple carbon sources are available and the microbial growth presents two or more growth phases [6]. The models also allow for the incorporation of proteomics data, and thus provide a framework for multi-omics data analysis and integration [3,7].

The parameters included in pcGEMs are: (i) the enzyme turnover numbers, $k_{cat}$, a first-order rate constant with the unit of $s^{-1}$, that describes the limiting rate of reactions catalysed by enzymes when these are fully occupied at their saturation point; and (ii) enzyme abundances (in mmol/gDW), obtained from quantitative proteomics experiments. Values of $k_{cat}$ can be measured from biochemical assays or estimated from computational methods based on constraint-based and data-driven approaches [8], while enzyme abundances are obtained from absolute proteomics measurements. More specifically, they are obtained from peptide intensity-based quantification or spectral counting [9]. However, proteomics experiments for absolute quantification are still difficult to perform, given the challenges put forward by the diversity of physicochemical properties of protein [10], lack of standards and problems in reproducibility [11], and overall inaccessibility given the high costs of equipment and supplies [12].

Computational methods have also been developed to predict protein abundance, mostly based on data-driven models. These models often explore the central dogma of molecular biology by assessing the relationship between transcription and protein biosynthesis. Notable approaches to estimate protein abundance include the joint learning approach devised by Li et al [13], where an ensemble model was constructed by combining different supervised learning algorithms, outperforming competing approaches in the NCI-CPTAC DREAM Proteogenomics Challenge. Another approach, developed by Terai and Asai [14], uses features such as the accessibility around the Shine-Dalgarno sequence, minimum free energy of the mRNA molecule, Viterbi score, and inside-outside score. Further, Ferreira et al. [15] explored codon usage bias information to train an AdaBoost regression model, achieving higher correlations than previous approaches without the usage of transcriptomics data.

Aside from machine learning models, constraint-based approaches have also been used to predict protein abundance. Using approaches such as MOMENT [2] or GECKO [3], it is possible to calculate the optimal concentration of enzymes necessary to carry the provided flux

with the provided catalytic rate, given the relationship:

$$v_j \leq k_{cat}^{ij} \cdot [E_i] \tag{1}$$

where $v_j$ is the metabolic flux of reaction $j$, $[E_i]$ is the concentration of an enzyme $i$, and $k_{cat}^{ij}$ is the catalytic rate of an enzyme $i$ catalyzing a reaction $j$. This allows for deriving $k_{cat}^{ij}$ values given the other two are available. This relationship was explored by Heckmann et al. [16] by using pcGEMs to predict enzyme concentrations given catalytic rates predicted computationally, achieving a 43% lower root mean squared error.

Assuming that pcGEMs that integrate proteomics data predict flux distributions that reflect the corresponding metabolic state, we ask whether the reverse operation could be employed to predict proteomics data that match a given physiological state. Moreover, as cells are exposed to stresses or changing environmental conditions, the current growth state is disturbed, leading to an alternative growth state in which gene expression, regulatory pathways and metabolic flux are changed in adjusting the cell to the new physiological condition [17]. Despite the aforementioned advances in predicting protein abundances, the problem of predicting enzyme allocation under alternative growth conditions remains largely unexplored. This could be useful to explore how cells adapt to changing environmental conditions, such as those faced by yeasts in industrial fermentations or by pathogenic bacteria when exposed to antibiotics.

Here we propose PARROT (Fig 1), for **P**rotein allocation **A**djustment fo**R** alte**R**native envi**rO**nmen**T**s, a family of constraint-based approaches for prediction of protein abundances for alternative growth conditions using protein abundances measured in a reference state. Our proposed approach is inspired by Minimization of Metabolic Adjustment (MOMA) [18], which minimizes the distance between a reference state and a gene knock-out state while ensuring cell survival in the later. We show that PARROT predicted enzyme concentrations in very good agreement with experimental data and outperformed competing methods for minimizing flux distributions. Therefore, PARROT can be used to parameterize pcGEMs for unseen, alternative growth conditions from which metabolic phenotypes can further be analysed.

## Results

### PARROT successfully captures protein allocation changes in yeast

We used PARROT to predict the enzyme usage distribution for 19 growth conditions under constraints provided by experimental data. First, we built a baseline for comparison with predictions from PARROT (Fig 1). To this end, we integrated the experimental proteomics measurements obtained from Lahtvee et al. [19], Yu et al. [20], Di Bartolomeo et al. [21], and Yu et al. [22] (S1 Table) in the ecYeast8 model and minimized the enzyme allocation (Methods). The resulting allocation of enzymes $\mathbf{E}_s^{exp}$ included 286 to 336 enzymes with abundance in all considered conditions. For the reference condition, we used the experimental proteomics measurements from the control growth conditions in the respective four groups of experiments, after flexibilization following GECKO 2.0 (see Methods) (S1 Table). The number of enzymes contained in $\mathbf{E}_{ref}$ ranged from 533 to 744, depending on the investigated control sample. With the resulting enzyme allocation at the reference and the baseline of an alternative growth condition, $\mathbf{E}_{ref}$ and $\mathbf{E}_s^{exp}$, we used four variants of PARROT (see Methods) to predict the enzyme allocation, $\mathbf{E}_s$, for the alternative growth condition.

The first variant of PARROT (referred as LP1) minimizes the Manhattan distance between $\mathbf{E}_{ref}$ and $\mathbf{E}_s$ (see Methods). The number of enzymes contained in the predicted $\mathbf{E}_s$ ranged from 224 to 253 over the considered experiments. When comparing the median of the calculated

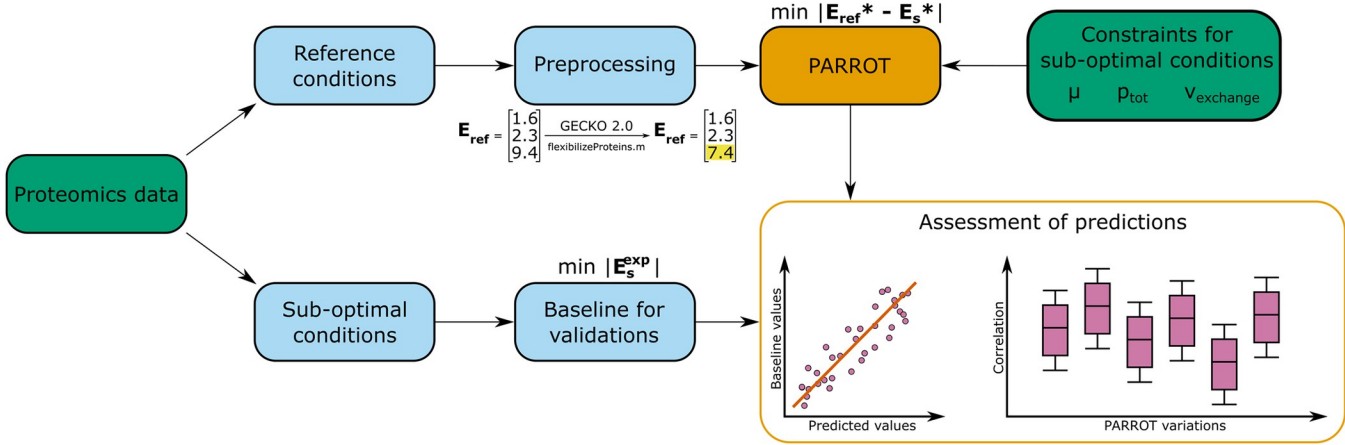

**Fig 1. Workflow of PARROT to predict enzyme usage for alternative growth conditions.** PARROT uses experimental proteomics data from a reference growth condition, and experimental physiological parameters from an alternative growth condition in a protein-constrained model. The proteomics data from the reference state is pre-processed by integrating the data in a pcGEM using the GECKO Toolbox 2 and allowing flexibility in its values. The proteomics data from the alternative state is used to generate a baseline, which is in turn used for comparison with predictions from the PARROT variants.

Pearson correlations between the baseline and predicted enzyme allocation correlations, we found that LP1 achieved a higher median correlation compared to pFBA and its modified implementation (Fig 2A). This variant also outperformed the null model, where $k_{cat}$ values are used directly as the enzyme usage ($E_{s,i} = k_{cat}^{ij}$). This negative control is useful to determine the contribution of $k_{cat}$ values to the correlation between $\mathbf{E_{exp}}$ and $\mathbf{E_s}$ (see Methods).

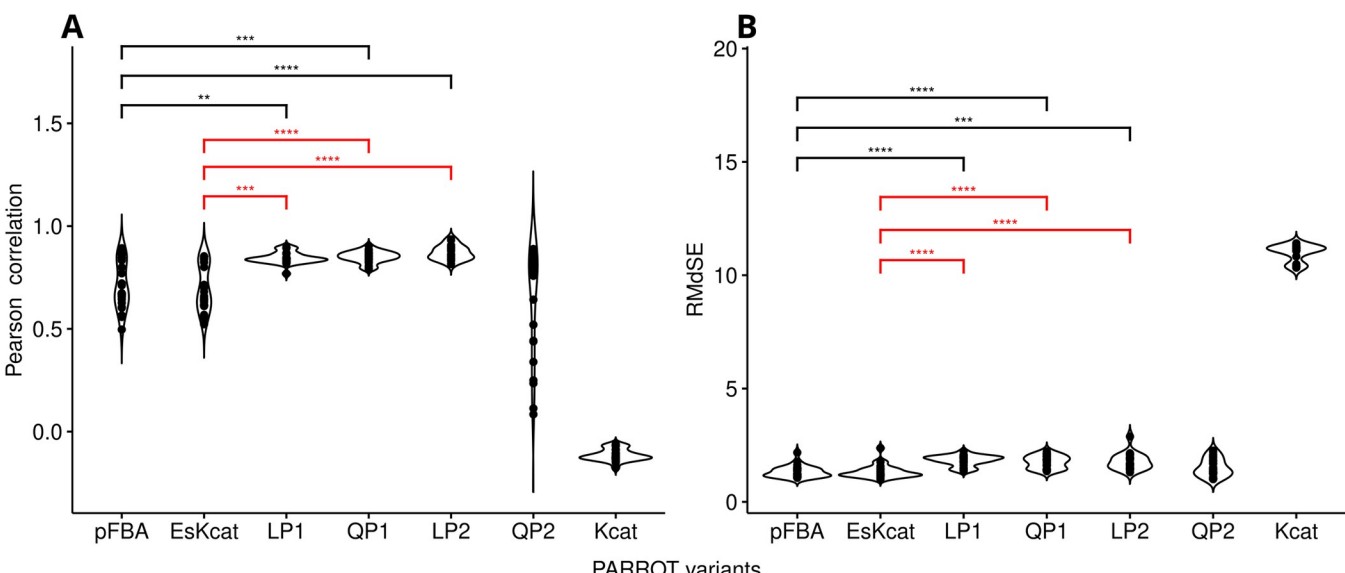

**Fig 2. Comparative performance analysis of PARROT with proteomics data from *S. cerevisiae*.** All protein abundance values were log10-transformed prior to comparisons. **a**. Pearson correlation calculated between predicted enzyme distribution and the baseline obtained from minimizing the first norm of the experimental enzyme usage distribution. The four variants of PARROT are denoted as LP1 (Manhattan distance of enzyme distributions), LP2 (weighted Manhattan distance, considering flux and enzyme distributions), QP1 (Euclidean distance of enzyme distributions), and QP2 (weighted Euclidean distance of flux and enzyme distributions). The performance of PARROT was compared to pFBA and its modified version EsKcat (first norm of enzyme usage), see Methods. A pairwise Wilcoxon rank sum assesses the statistical significance: **** p-value $< 1 \cdot 10^{-5}$, *** p-value $< 2 \cdot 10^{-4}$, ** p-value $< 5 \cdot 10^{-4}$. **b**. Assessment of model performance based on the root median squared error (RMdSE). A pairwise Wilcoxon rank sum assesses the statistical significance: **** p-value $< 9 \cdot 10^{-6}$, *** p-value $< 2 \cdot 10^{-5}$. Black significance bar indicates comparisons to pFBA. Red significance bar indicates comparison to EsKcat.

The second variant of PARROT, referred as QP1, minimizes the Euclidean distance between $E_{ref}$ and $E_s$ (see Methods). The predicted $E_s$ ranged from 210 to 258 predicted enzymes over the considered experiments. We found that QP1 achieved a higher median correlation when compared to pFBA and its modified implementation, when considering the median of the calculated Pearson correlations between the baseline and $E_s$ (Fig 2A). As with LP1, this variant outperformed the null model.

The third variant of PARROT, referred as LP2, minimizes the weighted sum of the Manhattan distance between enzyme usage distributions and the Manhattan distance between flux distributions. Thus, this variant also considers the metabolic fluxes of each condition along with the enzyme usage distribution. As observed for the other variants, LP2 outperformed pFBA and its modified implementation, when comparing the median of the calculated Pearson correlations between the baseline and $E_s$, while also outperforming the null model.

The fourth and final variant of PARROT, QP2, minimizes the weighted sum of the Euclidean distance between enzyme usage distributions and the Euclidean distance between flux distributions. Unlike other variants, QP2 did not achieve a higher median Pearson correlation when comparing the predictions to the baseline, but it was better than the null model. However, the root median squared error (RMdSE) between predictions and the baseline was the lowest among variants, being comparable to pFBA and its modified implementation (Fig 2B). Taken together, the results demonstrated that PARROT achieved good predictive performance based on the data from *S. cerevisiae* when compared to pFBA and its modified implementation.

## Different variants of PARROT outperformed the benchmarks for *E. coli*

To verify if the conclusions from PARROT hold in another unicellular model organism, we applied it to predict enzyme allocation $E_s$ in alternative growth conditions for *E. coli* given constraints provided by growth experiments. As in the case of *S. cerevisiae*, we built a baseline for comparison with the predictions obtained from PARROT by integrating the experimental proteomics measurements from Valgepea et al. [23], Peeno et al. [24] and Schmidt et al. [25] (S2 Table) in the eciML1515 model, and minimized the total enzyme allocation (see Methods). The resulting $E_s^{exp}$ included protein allocation for 164 to 176 enzymes. Further, as reference condition we considered the control samples or the chemostat measurements with the smallest dilution rate (S2 Table). We chose the smallest dilution rate to ensure that cells are growing aerobically and to prevent the metabolic shifts seen in higher dilution rates (e.g., overflow metabolism). The number of enzymes contained in $E_{ref}$ ranged from 152 to 188 depending on the control experimented used.

The prediction of $E_s$ distributions and their assessment were similar to *S. cerevisiae*. The LP1 variant predicted between 122 and 133 enzymes for $E_s$ across conditions. This variant exhibited significantly higher median correlations compared to pFBA (p-value = $1.24 \cdot 10^{-13}$ for Pearson correlations, pairwise Wilcoxon rank sum test) (Fig 3A). For the QP1 variant, the number of predicted enzymes ranged from 115 to 133 across conditions. Further, it had higher median correlations compared to pFBA, but lower than its modified implementation. The LP2 variant performed similar to LP1, predicting between 125 and 137 enzymes and achieving higher median correlations compared to pFBA and its modified implementation. The variant QP2, on the other hand, predicted a wider range and had a low number of enzymes, ranging from 19 to 141. This variant also had lower median correlations compared to pFBA and its modified implementation. As in *S. cerevisiae*, the QP2 variant had lower RMdSE errors than the other variants. All variants outperformed the null model in all comparisons. These findings demonstrated that PARROT is applicable with data from another microorganism without decrease in performance.

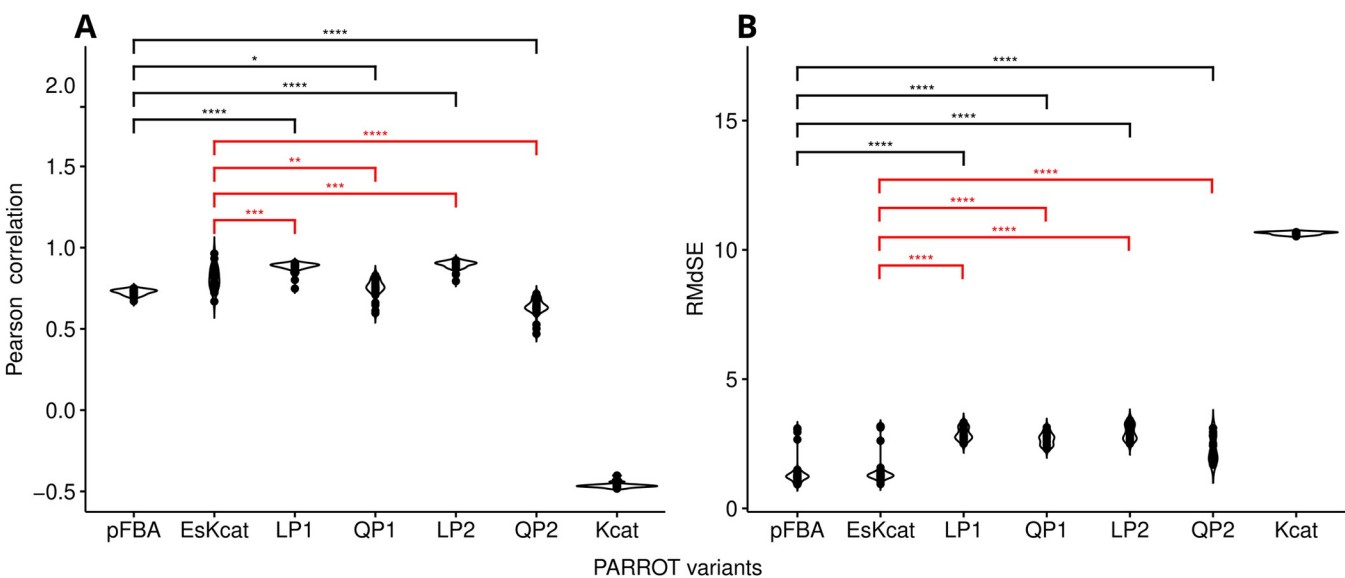

**Fig 3. Comparative performance analysis of PARROT with proteomics data from *E. coli*.** All protein abundance values were log10-transformed prior to comparisons. **a**. Pearson correlation calculated between predicted enzyme usage distribution and the baseline obtained from minimizing the first norm of the experimental enzyme usage distribution. A pairwise Wilcoxon rank sum assesses the statistical significance: **** p-value $< 2 \cdot 10^{-11}$, *** p-value $< 2 \cdot 10^{-4}$, ** p-value $< 6 \cdot 10^{-3}$, * p-value $< 3 \cdot 10^{-2}$. **b**. Assessment of model performance based on the RMdSE in *E. coli*. A pairwise Wilcoxon rank sum assesses the statistical significance: **** p-value $< 1 \cdot 10^{-5}$. Black significance bar indicates comparisons to pFBA. Red significance bar indicates comparison to EsKcat.

## Robustness analysis shows the consistency of prediction from PARROT

To further evaluate the predictions made by PARROT, we investigated if the construction of the baseline could impact the correlations. Thus, we reconstructed the baseline by minimizing the 2-norm of the vector $\mathbf{E}_s^{exp}$ instead of the 1-norm. To this end, we repeated all comparisons as performed for a baseline constructed by minimizing the 1-norm, using the predicted $\mathbf{E}_s$ obtained by the PARROT variants. Importantly, the results were consistent between the two baseline approaches. For *S. cerevisiae*, the LP2 variant achieved the highest mean Pearson correlations compared to pFBA and its modified implementation (S1 Fig). For the RMdSE, all PARROT variants had errors comparable to the positive controls (S2 Fig). As observed in the comparisons with the 1-norm baseline, all PARROT variants outperformed the null model.

The comparisons performed using predictions obtained for *E. coli* were also consistent with different variants of PARROT that outperformed pFBA. Considering the Pearson correlations, the LP1 and the LP2 variants also had the highest median correlations and were significantly different to pFBA. Likewise, these PARROT variants also showed significant difference to the modified implementation of pFBA (S3 Fig). The comparison of RMdSE values were also consistent with this observation, as the errors were comparable to the positive controls (S4 Fig). Altogether, these results highlight the robustness of estimations of $\mathbf{E}_s$ obtained from PARROT.

## Proteome-aware minimalization is more relevant than minimization of flux distances

Given that LP2 and QP2 make use of a weighting factor λ, we were interested in how different λ values impact the predictions. We used λ values ranging from 0 (no fluxes used) to 1 (fluxes and enzyme usages equally considered). We also considered a scenario of λ values ranging from 0.1 to 1 in order to probe different solutions where metabolic fluxes are always considered. We considered a λ value to be optimal if it resulted in the highest Pearson correlation to

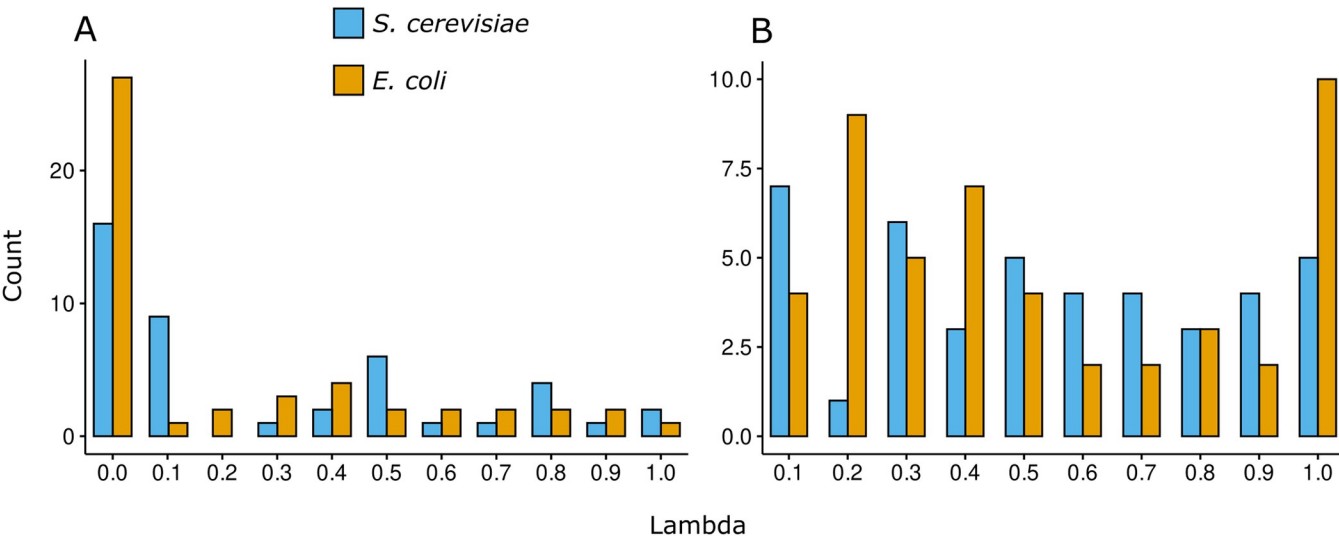

**Fig 4. Optimal λ values across conditions and PARROT variants.** The optima λ value was determined by optimising the LP2 and QP2 variants and finding the value that outputs predictions with the highest Pearson correlation when compared to the baseline. Blue bars correspond to *S. cerevisiae*, and orange bars correspond to *E. coli*. **a**. Number of occurrences of an optimal λ value in a range of 0 to 1. Note that a λ value of zero means that no fluxes are used for the objective, being equivalent to the LP1 and LP2 variants. **b**. Number of occurrences of an optimal λ value in a range of 0.1 to 1. In this scenario, fluxes are always used for the objective.

the baseline. In the first scenario, for both *S. cerevisiae* and *E. coli* the most frequent optimal λ was 0, with decreasing correlation values as λ values increased (Fig 4A). In the second scenario, the optimal λ values were more equally distributed, with *S. cerevisiae* having a higher frequency of lower values (Fig 4B). For *E. coli*, lower λ values were also frequent, while also having a λ of 1 slighly more frequent than a λ of 0.2 (Fig 4B). Taken together, these results indicate that the problem of minimizing enzyme usage contributes more to predictions than minimizing metabolic fluxes.

## Discussion

Here we proposed a family of constraint-based approaches, termed PARROT, that address the problem of predicting reallocation of protein abundance from a reference growth condition to an alternative growth condition. PARROT is based on the principle that organisms tend to minimally adjust cellular physiology between growth conditions to make effective use of resources [26]. The predictions of enzyme allocation generated by PARROT rely on quantitative proteomics data for a reference condition. The resulting optimization problems constructed are thus similar to MOMA, which depends on a model representing a wild-type strain to predict a minimally adjusted flux distribution for a mutant strain.

Understanding how cells adjust enzyme allocation during growth conditions apart from the physiological optimum might prove useful to study, for example, the adaptability of yeasts when exposed to ethanol during fermentation. Ethanol hinders growth and enacts several changes to membrane structure and function, causes protein denaturing and metabolic imbalances [27]. The yeast *Kluyveromyces marxianus*, important for fermentation of dairy products, adapts to ethanol stress by strengthening the cell membrane by accumulating trehalose and by altering the content of ergosterol and unsaturated fatty acids in the cell membrane, along with changes in several regulatory pathways [28,29]. Another example is the resistance to antibiotics observed in human and animal bacterial pathogens. The prolonged and continuous exposure to antibiotics leads to a selective pressure where the bacterial population acquires resistance to

the antibiotic which it is being exposed to, by means of mutations or through acquisition of mobile genetic elements, leading to the emergence of antibiotic-resistant strains [30]. For instance, antibiotic-resistant strains of *Klebsiella pneumoniae*, an ESKAPE pathogen, harbours many plasmids containing genes responsible for coding enzymes that break the antibiotic molecule. An example is the ampC gene family, which codes for β-lactamases, capable of degrading antibiotics such as penicillin, monobactams, cephalosporins and carbapenems. Resistance to these antibiotics is achieved by overexpression of these genes and overproduction of β-lactamases, along with the production of binding proteins that target the antibiotic molecules [31]. Overproduction of enzymes leads to disrupted metabolic states due to the inefficient allocation of resources [32], which can be exploited for therapeutic efforts such as enhancing antibiotic sensitivity [33].

By comparing the predictions to a baseline constructed with experimental proteomics measurements for alternative growth conditions, we found that PARROT predicted protein abundances with very good agreement with the baseline. In addition, we demonstrated that these predictions were consistent and robust to how the baseline is constructed. The performance of PARROT also holds for two model organisms, *S. cerevisiae* and *E. coli*, highlighting the general application of the principle of minimal protein adjustment on which the predictions are based.

From the different variants of PARROT, LP1 (minimization of the Manhattan distance of enzyme usage distributions) and LP2 (the minimization of the weighted sum of the Manhattan distance of enzyme usage and Manhattan distance of flux distributions) were the best contenders across conditions for both *S. cerevisiae* and *E. coli*. The variant QP1 (minimization of the Euclidean distance of enzyme usage distributions) resulted in good, but inconsistent performance between *S. cerevisiae* and *E. coli*. For QP2 (the minimization of the weighted sum of the Euclidean distance of enzyme usage and Euclidean distance of flux distributions), it had poor results for *S. cerevisiae*, while having good results for *E. coli*, albeit worse than the other variants. This agrees with the fact that the first norm distance is the natural metric for enzyme abundances in the cell, because a change in enzyme concentration requires ribosomal activity that scales linearly with the enzyme abundance [34].

The baseline approach devised to assess the predictions allows for a fair comparison between the predicted enzyme usage distribution and the experimental protein abundance values. In constraining the pcGEMs with the proteomics measurements, the experimental values are first readjusted to match the enzyme levels that actually carry flux in the model, since more protein is produced than actually needed by the cell [35]. This, however, implies that the predicted values are not directly comparable to experimental proteomics values, which affect the determined measures of performance. By adjusting the experimental values to levels that are compatible with what is actually employed to carry metabolic flux, we could more adequately assess the correlation with enzyme allocation predicted from the pcGEMs, albeit losing the direct correspondence to experimental data.

The parameter λ is a factor that weights the usage of metabolic fluxes for the optimisation problem. By varying this value between 0 and 1, we could assess how much the minimization of metabolic fluxes contributes to the problem of predicting enzyme usage. A λ value of 0 would render the variants LP2 and QP2 equivalent to LP1 and QP1, respectively, as metabolic flux would be neglected in the optimal solutions. A λ value of 1, in the other hand, renders LP2 and QP2 as equivalent to using a pcGEM with the canonical implementation of MOMA, which considers all fluxes equally. When the two PARROT variants are free to vary λ between 0 and 1, the optimum is reached for lower λ values. This can be explained by the experimental observation that in changing environments, cells adopt a strategy of initially adjusting gene expression, which subsequently results in shifts in protein allocation. Consequently, this leads to subsequent changes in metabolic flux [21,36]. When constraining λ to a value between 0.1

and 1, higher values of λ are present but still not more prevalent than lower values of λ. This suggests that the joint minimization of fluxes and enzymes is not a principle of flux redistribution, given that when higher λ values are employed in the objective function of LP2 and QP2, the model simultaneously optimizes both the enzyme usage distribution and the flux distribution, as though the cell performs these processes at the same time. Instead, the principle is guided by minimization of resource redistribution, as best captured by LP1 and QP1, and by LP2 and QP2 with low values of λ, from which a flux redistribution is then later derived. Using lower λ values, the model prioritizes minimizing the enzyme usage distribution, aligning more closely with experimental observations. With a λ of 0, the model disregards metabolic fluxes entirely, enabling it to focus on solving for minimal enzyme usage redistribution, only calculating the flux distribution as a function of the former, mirroring what is observed in the cell [37]. Thus, by being proteome-aware, PARROT is better suited for simulations using pcGEMs than the quadratic and linear implementations of MOMA, given that higher participation of metabolic fluxes lowers the overall predictive performance. Altogether, we demonstrated that minimizing the readjustment of enzyme resource allocation is one principle underpinning microbial adjustment to an alternative condition, aligning with experimental evidence. Thus, PARROT may allow for study and engineering of microbial cell factories, as these are often under suboptimal growth conditions in industrial settings [38].

Despite the advantages of using a baseline, predictions of enzyme levels using Eq (1) still underestimates protein abundance, leading to a disparity between predictions and *in vivo* concentrations. This remaining portion of proteins, termed the "proteome reserve", is useful for the cell to quickly adapt to unstable environments, being an evolutionary conserved strategy [39]. It is important to highlight, though, that this reasoning does not assume that cells are operating at the saturation point for all metabolites, but rather that enzymes are used inefficiently. If enzymes are operating near $V_{max}$, then enzymes would be the only cellular components that exert control on metabolic fluxes. As noted by [40], however, cell overexpress enzymes and uses metabolite concentrations to control metabolic flux. This falls in line with the evolutionary conservation of protein stoichiometries at the pathway level as demonstrated by Lalanne et al. [41]. Although it is still not understood how preferred enzyme stoichiometry is determined, it was observed that the preferred range of enzyme stoichiometry follows a narrow distribution among pathways in Gram-positive and -negative bacteria, likely a result of evolutionary conservation or convergence. As suggested in the study, protein biosynthesis and consequently its usage is bound to a cost-benefit trade-off, where the optimal level of enzymes is balanced with the need for a buffer zone in case of changing environments. Similar to our approach, the works of Mori et al. [39] and Lalanne et al. [41] deals with proteome reallocation in an alternative growth condition. However, the first deals with proteome sectors, while the latter concerns with pathway-centric stoichiometries. Our approach thus differs as we consider protein reallocation for each enzyme individually.

Nevertheless, other approaches for estimating *in vivo* protein concentrations would still need to overcome the underestimation of protein concentrations of pcGEMs, especially by considering the proteome reserve. Interestingly, Alter et al. [42] deal with the problem of catalytically inactive enzymes by considering a protein sector of unused enzymes, along with active enzymes sector and the translational protein sectors. The unused protein sector, $\phi_{UE}$, is calculated by relating the decrease of the concentration of unused enzymes to the increase in the substrate uptake rate. This allows for the model to predict the adaptability of *E. coli* to changing environmental conditions. However, the initial value $\phi_{UE,0}$, which is the unused enzyme concentration when the substrate uptake is zero, is obtained from simulations using a ME model, which are notoriously difficult to parameterize [43]. New approaches could include features such as cellular machinery beyond enzymes that participate in metabolism, or by integrating

constraint-based approaches with data-driven approaches, as recently done in the CAMEL approach [44].

## Material and methods

### The principle of minimizing the change in enzyme usage between an alternative and reference state

To find the enzyme distribution vector that matches the enzyme usage of a cell growing in alternative growth conditions, we propose PARROT, an approach that minimizes the distance between a reference enzyme allocation $\mathbf{E_{ref}}$ and an alternative growth enzyme allocation $\mathbf{E_s}$ (Fig 1). This is consistent with observations that micro-organisms minimize expenditures to perform growth and to maintain the associated flux state [26]. We define and compare four different objectives to model the distance between the allocation of enzymes in alternative and reference growth states: (i) the Manhattan distance; (ii) the Euclidean distance; (iii) the weighted sum of the Manhattan distance between enzyme allocations and the Manhattan distance between flux distributions; (iv) the weighted sum of the Euclidean distance between enzyme allocations and the Euclidean distance between flux distributions. The first can be formulated as a linear optimization problem (LP1), specified as follows:

$$\min \| \frac{\mathbf{E_{ref}}}{E_{ref}^{tot}} - \frac{\mathbf{E_s}}{E_s^{tot}} \|_1 \tag{2}$$

$$\text{s.t. } \mathbf{Nv} = \mathbf{0} \tag{3}$$

$$v_{s,min} \leq v_s \leq v_{s,max} \tag{4}$$

$$v_s \leq k_{cat} \cdot [E_s] \tag{5}$$

$$\sum E_s = E_s^{tot} \tag{6}$$

$$v_{bio} = \mu, \tag{7}$$

where $E_{ref}^{tot}$ and $E_s^{tot}$ represent the total enzyme usage in the model for the reference and alternative states, respectively; $\mathbf{N}$ is the stoichiometric matrix; $\mathbf{v}$ is the flux distribution vector; $v_{bio}$ is the flux through the biomass pseudo-reaction; and $\mu$ is the specific growth rate, determined from measurements in the alternative state. The other objectives are captured by the following:

$$\text{QP1}: \| \frac{\mathbf{E_{ref}}}{E_{ref}^{tot}} - \frac{\mathbf{E_s}}{E_s^{tot}} \|_2, \tag{8}$$

$$\text{LP2}: \| \frac{\mathbf{E_{ref}}}{E_{ref}^{tot}} - \frac{\mathbf{E_s}}{E_s^{tot}} \|_1 + \lambda \| \mathbf{v_{ref}} - \mathbf{v_s} \|_1, \tag{9}$$

$$\text{QP2}: \| \frac{\mathbf{E_{ref}}}{E_{ref}^{tot}} - \frac{\mathbf{E_s}}{E_s^{tot}} \|_2 + \lambda \| \mathbf{v_{ref}} - \mathbf{v_s} \|_2. \tag{10}$$

where the parameter λ is a weighting factor chosen by inspecting the difference between the norms of enzyme allocation and the flux distributions. We solved the corresponding problems under the same constraints as in Eq 2. We implemented and solved the problems in MATLAB (The MathWorks Inc., Natick, Massachusetts) using the COBRA Toolbox [45] and the Gurobi

solver v9.1.1 [46]. The implementation of PARROT can be found in the GitHub repository: https://github.com/mauricioamf/PARROT.

## Experimental data and simulation constraints

To test the variants of the proposed approach, PARROT, we used the pcGEMs of *Saccharomyces cerevisiae*, ecYeast8 [47], and *Escherichia coli*, eciML1515 [48]. We employed quantitative proteomics measurements for both species performed in a number of growth conditions, ranging from optimal growth in standard physiological conditions to stress conditions, alternative nutrient usage and chemostat cultivation.

For *S. cerevisiae*, we used the protein measurements from Chen and Nielsen [49] for 19 different growth conditions, which were collected from four studies [19–22]. These included proteomics measurements in yeast growing in ethanol, osmolarity, and high temperature stresses [19]; yeast growing in chemostats with reducing nitrogen availability [20]; and yeast growing in chemostats limited by the nitrogen source in increasing dilution rates and in chemostats with alternative nitrogen sources [22]. We also used measurements of nutrient uptake rates, growth rates and protein content from these studies to constrain the batch model, which does not consider protein measurements and rely on the protein pool constraint.

For *E. coli*, we used the proteomics data for 20 different growth conditions collected in [50] from three different studies [23–25]. These include batch cultivations of *E. coli* growing with different carbon sources and a glucose-limited chemostat culture, with dilution rates ranging from 0.12 $h^{-1}$ to 0.5 $h^{-1}$ performed by Schmidt et al. [25], a second chemostat limited by glucose at dilution rates ranging from 0.11 $h^{-1}$ to 0.49 $h^{-1}$ [23], and a third chemostat limited by glucose at dilutions rates ranging from 0.21 $h^{-1}$ to 0.51 $h^{-1}$ [24]. Similar to *S. cerevisiae*, the batch model was constrained with the nutrient uptake rates, growth rates and protein content measured in the studies where the protein measurements were taken. For both species, we excluded the conditions that did not have measured uptake rates, growth rates, or protein content. In addition, we excluded the temperature stress conditions from Lahtvee et al. [19], as temperature can severely impact the function of enzymes [51], and temperature stress responses entail changes beyond metabolic flux redistribution [17]. To prevent overconstraining the models, we allowed 5% flexibility on the growth rate. For other constraints, we allowed flexibility by increments of 1% until the measured growth rate is achieved.

## Pre-processing of protein measurements for the reference state

From the protein measurements obtained from Davidi et al. [50] and Chen and Nielsen [49] we separated the measurements according to each experiment performed in the original studies. From each experiment we selected the control sample to represent the reference state in our approach PARROT. We corrected the protein measurements for the reference state measurements by integrating the values into the pcGEMs ecYeast8 and eciML1515 for *S. cerevisiae* and *E. coli*, respectively, using the GECKO Toolbox 2 [48]. The GECKO Toolbox 2 identifies the enzyme usage values that most limit growth and allow flexibility in the enzyme usage constraints to prevent over-constraining the model. For the $E_{ref}$ vector of each experiment, we then used these relaxed protein measurements, while keeping the original values for proteins that were unchanged by allowing flexibility.

## Assessment of predicted enzyme usage distributions

The protein measurements, $E_s^{exp}$, for the alternative growth conditions obtained from Davidi et al. [50] and Chen and Nielsen [49] were not used directly in simulations. These experimental measurements were instead employed to calculate a baseline to which predictions of $E_s$

were compared. Assuming that simulations performed with pcGEMs use only the optimal concentration of enzymes necessary to carry a given metabolic flux, the model-allocated protein usage would underestimate the *in vivo* enzyme concentrations. To allow for a fair comparison, we devised a baseline by integrating the experimental proteomics measurements of each experiment into the pcGEMs using the GECKO Toolbox 2 in which we minimized the total enzyme allocation given the following optimization problem:

$$\min\|\mathbf{E}_\mathbf{s}^\mathbf{exp}\|_1 \tag{11}$$

$$\text{s.t. } \mathbf{Nv} = \mathbf{0} \tag{12}$$

$$\mathbf{v}_\mathbf{s,min} \leq \mathbf{v}_\mathbf{s} \leq \mathbf{v}_\mathbf{s,max} \tag{13}$$

$$v_{s,j} \leq k_{cat}^{ij} \cdot [E_s^{exp,i}] \tag{14}$$

$$\sum E_s^{exp} = E_s^{exp,tot} \tag{15}$$

$$v_{bio} = \mu. \tag{16}$$

The resulting enzyme usage distribution, $\mathbf{E}_\mathbf{s}^\mathbf{exp}$, was then defined as the baseline for each sample of each proteomics experiment. We compared the predicted $\mathbf{E}_\mathbf{s}$ values from the four variants of PARROT to $\mathbf{E}_\mathbf{s}^\mathbf{exp}$ by calculating the Pearson correlations of each sample. Further, we calculated the root-median square error (RMdSE) to measure the difference between predicted and baseline values. For assessing both correlations and the RMdSE, we log10-transformed the values for the predictions and the baseline.

We also performed a robustness analysis by checking the effect of minimizing the 2-norm of $\mathbf{E}_\mathbf{s}^\mathbf{exp}$ to construct the baseline, instead of the 1-norm, keeping the constraints defined in Eqs 12–16:

$$\min\|\mathbf{E}_\mathbf{s}^\mathbf{exp}\|_2 \tag{17}$$

We compared the predictions of our approaches to those obtained using an extension of parsimonious enzyme usage FBA (pFBA) [52] to consider enzyme constraints. It is suitable for benchmarking our approach given its ability to predict phenotypes in good accordance with other methods that rely on transcriptomics and proteomics data. To this end, for each sample of each experiment, we defined the optimization problem as:

$$\min \sum_{j=1}^m v_{j,s,irrev} \tag{18}$$

$$\text{s.t. } \mathbf{N}_\mathbf{s,irrev} \cdot \mathbf{v}_\mathbf{s,irrev} = \mathbf{0} \tag{19}$$

$$0 \leq \mathbf{v}_\mathbf{s,irrev} \leq \mathbf{v}_\mathbf{s,irrev,max} \tag{20}$$

$$v_{s,irrev,j} \leq k_{cat}^{ij} \cdot [E_{s,i}] \tag{21}$$

$$\sum E_s = E_s^{tot} \tag{22}$$

$$v_{bio} = \mu, \tag{23}$$

where $v_{j,s,irrev}$ corresponds to the flux distribution of an irreversible model in an alternative growth condition. We also assessed a modified version of pFBA with enzyme constraints with the following objective:

$$\min \sum\nolimits_{j=1}^{m} E_{s,i} \cdot k_{cat}^{ij}. \tag{24}$$

For pFBA and the modified implementation, we applied the same constraints on nutrient uptake rates and growth rates as for the four approaches assessed previously, and calculated the Pearson correlations and the RMdSE. Lastly, as a negative control to benchmark the performance of PARROT, we equated $E_{s,i}$ to $k_{cat}^{ij}$ ($E_{s,i} = k_{cat}^{ij}$), meaning that $k_{cat}$ values we used directly as the enzyme usage. This allows for determining how much of the correlation between $\mathbf{E_{exp}}$ and $\mathbf{E_s}$ can be attributed directly to $k_{cat}$ values. We calculated the correlation values and RMdSE for all assessed optimization problems and compared them to the predictions of pFBA and its modified implementation using a Pairwise Wilcoxon rank sum test with Bonferroni correction.

## Assessment of optimal values for the λ weighting factor

To systematically assess the impact of different lambda values, we optimised the LP2 and QP2 variants using λ values ranging from 0 (no fluxes used) to 1 (fluxes and enzyme usages equally considered). Additionally, we optimised the LP2 and QP2 variants using λ values ranging from 0.1 to 1 in order to make sure fluxes are always used for the objective function. In both scenarios, we calculated the Pearson correlation to the baseline for each λ value. We determined the optimal λ value as the value that outputs predictions with the highest Pearson correlation when compared to the first norm baseline.

## Supporting information

**S1 Table. Experimental proteomics measurements used for yeast.**
(DOCX)

**S2 Table. Experimental proteomics measurements used for *Escherichia coli*.**
(DOCX)

**S1 Fig. Pearson correlation calculated between predicted enzyme distribution and the baseline obtained from minimizing the 2-norm of the experimental enzyme usage distribution, in *S. cerevisiae*.** All values were log10-transformed prior to comparisons. A pairwise Wilcoxon rank sum assesses the statistical significance: ** p-value < 0.0009. Black significance bar indicates comparisons to pFBA. Red significance bar indicates comparisons to EsKcat.
(TIFF)

**S2 Fig. Assessment of model performance based on the root median squared error (RMdSE).** The minimization of the 2-norm of the experimental enzyme usage distribution in *S. cerevisiae* was used. All values were log10-transformed prior to comparisons.
(TIFF)

**S3 Fig. Pearson correlation calculated between predicted enzyme distribution and the baseline obtained from minimizing the 2-norm of the experimental enzyme usage distribution, in *E. coli*.** All values were log10-transformed prior to comparisons. A pairwise Wilcoxon rank sum assesses the statistical significance: **** p-value < 0.000005, * p-value < 0.03. Black significance bar indicates comparisons to pFBA. Red significance bar indicates

comparisons to EsKcat.
(TIFF)

**S4 Fig. Assessment of model performance based on the root median squared error (RMdSE).** The minimization of the second norm of the experimental enzyme usage distribution in *E. coli* was used. All values were log10-transformed prior to comparisons.
(TIFF)

## Acknowledgments

We thank Marius Arend, Philipp Wendering and Eduardo Almeida for their critical discussion and comments on this study.

## Author Contributions

**Conceptualization:** Mauricio Alexander de Moura Ferreira, Wendel Batista da Silveira, Zoran Nikoloski.

**Funding acquisition:** Wendel Batista da Silveira, Zoran Nikoloski.

**Investigation:** Mauricio Alexander de Moura Ferreira.

**Methodology:** Mauricio Alexander de Moura Ferreira.

**Project administration:** Wendel Batista da Silveira, Zoran Nikoloski.

**Software:** Mauricio Alexander de Moura Ferreira.

**Supervision:** Zoran Nikoloski.

**Validation:** Mauricio Alexander de Moura Ferreira.

**Writing – original draft:** Mauricio Alexander de Moura Ferreira, Wendel Batista da Silveira, Zoran Nikoloski.

**Writing – review & editing:** Mauricio Alexander de Moura Ferreira, Wendel Batista da Silveira, Zoran Nikoloski.

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
