## [Decision Letter · Decision Letter 0]

20 Jun 2023

Dear Dr. Nikoloski,

Thank you very much for submitting your manuscript "PARROT: Prediction of enzyme abundances using protein constrained metabolic models" for consideration at PLOS Computational Biology.

As with all papers reviewed by the journal, your manuscript was reviewed by members of the editorial board and by several independent reviewers. In light of the reviews (below this email), we would like to invite the resubmission of a significantly-revised version that takes into account the reviewers' comments.

In particular, the reviewers suggest edits to better clarify the results and substantiate your claims.

We cannot make any decision about publication until we have seen the revised manuscript and your response to the reviewers' comments. Your revised manuscript is also likely to be sent to reviewers for further evaluation.

Sincerely,

Christos A. Ouzounis

Academic Editor

PLOS Computational Biology

Stacey Finley

Section Editor

PLOS Computational Biology

Reviewer's Responses to Questions

**Comments to the Authors:**

Reviewer #1: The authors present their method, PARROT, which employs constraint-based approaches for minimizing the enzyme allocations to assess the protein allocations in microorganisms growing under different suboptimal conditions. These constraint-based approaches attempt at optimizing the enzyme allocations rather than other contending approaches which rely on flux redistributions. The authors benchmarked the performance of their approach/es on two model microorganisms---E. coli and S. cerevisiae---and propose the use of protein-constrained genome-scale metabolic models (pcGEMs) for studying microbes in different conditions.

Major Issues:

1. Given that the growth of microorganisms in different conditions, particularly suboptimal conditions, is central to the positioning of this method and the manuscript, it is important that the authors provide some more motivation and context for the need to study the microbes is such varied conditions.

The authors can easily find suitable literature from antimicrobial resistance where microbes, like the superbug ESKAPE pathogens, evolve and continue to grow even under the effect of antibiotics---the suboptimal growth conditions. There are studies that have reported evidences of a microbial strain becoming resistant to an antibiotic under prolonged exposures.

- doi.org/10.3389/fmicb.2021.714284

- doi.org/10.1093/femsre/fux013

Another avenue for reference literature could be the clinical growth of microbial communities using selective enrichment media.

2. The Results section should be rewritten to document the specifics and performance details of all approaches being compared/benchmarked. There are subsections titling PARROT to have outperformed other contending methods, but this is an inference to be made from the specific results. The exact specifics about the differences and similarities in results from different approaches are either not well documented/highlighted or are mentioned in the supplementary material. Therefore, from a reader's point of view, the actual results and performance of PARROT are not easy to follow and interpret.

Reviewer #2: PARROT: Prediction of enzyme abundances using protein constrained metabolic models

Summary:

This manuscript introduces a method for calculating Protein allocation Adjustment foR suboptimal enviROnmenTs (PARROT). With this method, the authors aim to study the metabolic phenotype of new, suboptimal conditions. The method is based on the principle of minimal adjustment of the proteome starting from a reference state. This is represented in the objective function which minimizes the distance between the reference condition and the suboptimal conditions. The authors compared different distance metrics (Manhattan and Euclidean distance) and the effect of adding the minimization of the flux distance to the objective functions. They found that for models of Saccharomyces cerevisiae and Escherichia coli, their method outperformed normal pFBA (minimizing the sum of all fluxes). Furthermore, they observed that the adding minimization of difference in flux distributions does not improve the results, even worsens it.

Major comments

General:

• Variants are used with different intentions (different conditions or different distance functions). This is confusing for the reader. Please be concise in how you refer to the different models or distance functions

Abstract:

• L. 21-22: In this sentence, you try to explain the main function of PARROT. I think this sentence is confusing for someone who hasn’t read the paper yet, since the meaning of ‘enzyme allocation’ and ‘enzyme allocation adjustment’ is unclear from the first part of the abstract. I recommend clarifying these terms or rephrasing to something a lay reader can understand.

• L 25: From the abstract it is unclear between what the distance is calculated. This could be easily added by replacing ‘of enzyme allocations’ with ‘between the allocation of enzymes of a reference and suboptimal condition’

Materials and methods:

• In the code 5% flexibility is allowed on the growth rate, which is not described in the Materials and methods.

• L. 357: It is not clear if 'values' refer to the original protein measurements or to the relaxed protein values.

• L. 382: What does ‘The second norm in constructing a baseline’ mean in mathematical terms? This is not clear from the text and is important in interpreting this analysis.

Results:

• L. 132: ‘18-336’ and L. 156: ‘19-141’. Why is the range of number of enzymes in the predicted suboptimal conditions so wide? How does this influence the interpretation of the results? I can imagine that the number of predictions influence the value of the accuracy metrics.

• L. 140-141: Here you mention that the kcat values are used as direct measure of the enzyme usage. What does this mean? Do you mean that you use the values of the enzyme variables as enzyme concentrations? I miss a description of this validation method in the ‘Methods’ section.

• l. 153: Why did you choose the smallest dilution rate as the reference condition? What is the motivation for this?

• Figure 2, 3, S2 and S3: These type of plots do not give a FAIR visualization of the results. Due to the scale of the plot, the distribution of the data in each box is impaired. I suggest adjusting the scale and trying out different ways of data visualization, such as a violin plot with individual data-points or even a raincloud plot.

• L. 189: Consistent with what? You can clarify this by adding ‘with this observation’ after ‘consistent’.

Discussion:

• L. 199-202 and 241-255: In the Results you describe the effect of alternating lambda on the occurrence of higher/lower Pearson correlation values. Later, in the Discussion you explain the meaning of a lambda value of 1 or 0. This part contains sufficient technical explanations, but I miss the biological explanation of this factor. What mechanism does this represent? What did you learn about Biology from this result?

• L. 280: ‘underestimating capacity’ I think capacity is not the correct word here. Capacity is a feature, not a result. I would use ‘the underestimation of protein concentrations’ instead.

• L. 281: Another approach is to specifically consider un/under-used enzymes as a separate protein sector as done in Alter et al. (2021) (https://doi.org/10.1128/msystems.00625-20). Consider discussing/citing this source.

Additional minor comments:

General: the use of ‘suboptimal‘ as term for the condition which is not the reference condition is questionable. For example, in the chemostat cultures, you choose the lower growth rate as reference condition, making the other conditions thus ‘suboptimal‘. In terms of biology this is incorrect: the cells are growing even faster in these conditions. I would choose a word as ‘alternative‘ condition instead.

l. 21, l. 294: I think the term ‘enzyme allocations’ is confusing, as allocation is not commonly used in plural. Perhaps 'the allocation of enzymes' suits better here?

l. 178: ‘than’ should be ‘compared to’

l.180: The following sentence: ‘As observed for comparisons using the first norm baseline’ could be replaced by: ‘As observed in the comparisons with the first norm baseline’. In my opinion, this would clarify the message of what is compared with what.

l. 233: ‘using’ should be ‘with’

l. 266-267: ‘is that cell overexpress’ could be replaced by ‘cells overexpress’

l. 293: ‘associated flux state’ A flux state cannot be performed, as it is a state. You can replace part of this sentence with: ‘to perform growth and to maintain the associated flux state’ or ‘for growth and the associated flux state’

l. 355, 357, 574: 'to flexibilise’ is not an English verb. You can replace this by ‘to allow flexibility in the enzyme usage constraints' or 'relaxing the enzyme usage constraints'.

Code on GitHub:

`ptotREF = 0.61;

f_REF = 0.4;

sigma_REF = 0.4;

f_STR = 0.4;

sigma_STR = 0.4;`

Where do these values originate from and what do they represent? Please include comments or documentation to interpret these values. What is the source of these numbers (or the assumptions made to define them)?

Using a dummy example on for example a core model to get to know the programming interface would be appreciated.

Reviewer #3: See attachment. I've left the original .odt file there in case you want to put your replies directly there.

**Have the authors made all data and (if applicable) computational code underlying the findings in their manuscript fully available?**

Reviewer #1: Yes

Reviewer #2: Yes

Reviewer #3: Yes

PLOS authors have the option to publish the peer review history of their article (what does this mean?). If published, this will include your full peer review and any attached files.

Reviewer #1: **Yes: **Kiran Gajanan Javkar

Reviewer #2: No

Reviewer #3: No
---

## [Decision Letter · Decision Letter 1]

5 Sep 2023

Dear Dr. Nikoloski,

Thank you very much for submitting your manuscript "PARROT: Prediction of enzyme abundances using protein-constrained metabolic models" for consideration at PLOS Computational Biology. As with all papers reviewed by the journal, your manuscript was reviewed by members of the editorial board and by several independent reviewers. The reviewers appreciated the attention to an important topic. Based on the reviews, we are likely to accept this manuscript for publication, providing that you modify the manuscript according to the review recommendations.

The reviewers have recognized the effort put into the revised submission, but raised some minor comments. We believe these points can be addressed in an updated submission.

Sincerely,

Christos A. Ouzounis

Associate Editor

PLOS Computational Biology

Stacey Finley

Section Editor

PLOS Computational Biology

Reviewer's Responses to Questions

**Comments to the Authors:**

Reviewer #1: I appreciate the authors for their efforts in updating the manuscript to incorporate the reviewers' feedback. I believe it has improved the manuscript substantially.

Having said that, there are still a few concerns that need to be addressed before this manuscript is accepted for publication:

1. Following the mentions from the title and the abstract section, the initial paragraphs of the introduction section should describe the need and utility of enzyme abundances and protein allocation, followed by subsequent literature on the challenges and advances in doing so. The current introduction starts with GEMs and protein-constraint GEMs and does not mention much about the need for enzyme abundances/protein allocations, with an underlying assumption that the reader already knows about the latter. It is recommended that the authors rethink and rewrite the initial paragraphs of the Introduction such that the context and scope of the manuscript are set for the readers without such major assumptions on the reader's part.

Additionally, some sentences have been worded in a bit convoluted manner while some other sentence combinations do not have a grammatical concordance that is easier to follow. For instance, the second sentence of the Abstract reads "Therefore, different experimental and machine learning approaches have been developed to quantify and predict protein abundances, respectively". The sentence structure suggests that it is contextual reasoning provided for what it described in the previous sentence. However, the previous sentence describes 'protein allocation', which is not mentioned in the following sentence at all. Such paraphrasing makes it quite difficult for a reader to follow the text. It is recommended that the authors proofread their manuscript with an end-reader in mind, preferably reviewing it with an English language expert, so that the presented content is easier to follow.

2. Line 101-102: "...the optimal growth state is disturbed". The authors need to define/describe what they refer to as the optimal growth state and the associated disturbance to it.

3. In the results section, the authors provide a count range of the enzymes by each model and indicate that since this count is similar to the experimental count, their model is performing better than the counterpart (pFBA). However, it is unclear how are count ranges alone sufficient to ascertain a better performance. Shouldn't the actual enzymes, themselves, be matching or similar? Alongside, the associated protein allocation counts also need to be accounted for, isn't it?

4. In lines 177--180, the authors mention their reasoning about their choice of the dilution rate while accounting for aerobic growth and metabolic shift prevention. This also raises a concern about the operational ranges for each of the 4 prediction models. The authors should describe the conditions that need to be satisfied for their proposed model/s to be suitable for these predictions.

Reviewer #2: Thank you for properly revising your manuscript. The adaptions and changes you made are adequate and do sufficiently consider the former concerns.

Reviewer #3: I thank the authors for considering my comments, as well as the comments from other reviewers. I am satisfied with the answers, and recommend to accept the manuscript.

**Have the authors made all data and (if applicable) computational code underlying the findings in their manuscript fully available?**

Reviewer #1: Yes

Reviewer #2: Yes

Reviewer #3: Yes

PLOS authors have the option to publish the peer review history of their article (what does this mean?). If published, this will include your full peer review and any attached files.

Reviewer #1: No

Reviewer #2: No

Reviewer #3: No

Figure Files:

Data Requirements:

Reproducibility:

References:

---

## [Decision Letter · Decision Letter 2]

29 Sep 2023

Dear Dr. Nikoloski,

We are pleased to inform you that your manuscript 'PARROT: Prediction of enzyme abundances using protein-constrained metabolic models' has been provisionally accepted for publication in PLOS Computational Biology.

Best regards,

Christos A. Ouzounis

Academic Editor

PLOS Computational Biology

Stacey Finley

Section Editor

PLOS Computational Biology

Reviewer's Responses to Questions

**Comments to the Authors:**

Reviewer #1: I thanks the authors for incorporating the feedback provided with my earlier review. I understand the authors' constraints or limitations with respect to the results benchmarking, but would sincerely like to have a better results benchmarking beyond just showing a similarity in the counts from two approaches.

Having said that, I am largely satisfied with the current manuscript, which I believe to be much improved than the earlier submission/s.

**Have the authors made all data and (if applicable) computational code underlying the findings in their manuscript fully available?**

Reviewer #1: None

PLOS authors have the option to publish the peer review history of their article (what does this mean?). If published, this will include your full peer review and any attached files.

Reviewer #1: **Yes: **Kiran Gajanan Javkar

---

## [Editor Report · Acceptance letter]

14 Oct 2023

PCOMPBIOL-D-23-00821R2 

PARROT: Prediction of enzyme abundances using protein-constrained metabolic models

Dear Dr Nikoloski,

I am pleased to inform you that your manuscript has been formally accepted for publication in PLOS Computational Biology. Your manuscript is now with our production department and you will be notified of the publication date in due course.

With kind regards,

Zsofia Freund
